# Electrochemical Detection and Characterization of Nanoparticles with Printed Devices

**DOI:** 10.3390/bios9020047

**Published:** 2019-03-28

**Authors:** Daniel Martín-Yerga

**Affiliations:** Department of Chemical Engineering, KTH Royal Institute of Technology, 100-44 Stockholm, Sweden; dmy@kth.se

**Keywords:** electrochemistry, printed electrodes, nanoparticles, electrochemical detection, low-cost analytical devices, biosensing, environmental analysis

## Abstract

Innovative methods to achieve the user-friendly, quick, and highly sensitive detection of nanomaterials are urgently needed. Nanomaterials have increased importance in commercial products, and there are concerns about the potential risk that they entail for the environment. In addition, detection of nanomaterials can be a highly valuable tool in many applications, such as biosensing. Electrochemical methods using disposable, low-cost, printed electrodes provide excellent analytical performance for the detection of a wide set of nanomaterials. In this review, the foundations and latest advances of several electrochemical strategies for the detection of nanoparticles using cost-effective printed devices are introduced. These strategies will equip the experimentalist with an extensive toolbox for the detection of nanoparticles of different chemical nature and possible applications ranging from quality control to environmental analysis and biosensing.

## 1. Introduction

Nanotechnology has revolutionized the landscape of modern technological and scientific development. Nanomaterials may have unique physical and chemical properties [1,2], such as an exceptionally high surface/volume ratio, confined electrons or enhanced biocompatibility compared to the same material at the macroscale. These particular properties can be exploited in innovative and valuable commercial products and specialized applications. Several types of nanomaterials are already being used in commercial products, such as silver nanoparticles (AgNPs), due to its antimicrobial and antifungal activity [3]. Quantum dots (QDs), semiconductor nanocrystals are present in displays of commercial devices exploiting their exceptional photoemission properties [4]. Perovskites seem like an excellent option for increasing the efficiency of solar cells [5]. Graphene could also make a significant impact as a building block of many functional materials [6] and technological products in the coming years. These are only a few examples of useful nanomaterials and their possible current or future commercial relevance. In order to have valuable industrial and commercial products, nanomaterials with particular chemical, physical, and mechanical properties are required. However, these properties greatly depend on nanoscale features, such as the size, geometry, surface charge or chemical modifications. Therefore, the importance of having suitable quality control processes for the characterization of synthesized nanomaterials is evident, and this will only increase in the coming years when more widespread use of these materials is expected. Additionally, all these new nanomaterials may end as waste, after their proper use, and reach the environment. There are concerns about the potential risk that many of these engineered nanomaterials could cause to human health or the environment [7]. Therefore, the development of analytical methods for the detection and characterization of engineered nanomaterials is a constant concern in the scientific world [8,9] also for environmental reasons. In this case, as relevant as the analytical quantification is the determination of other properties such as the size, since the biotoxicity could also depend on it [10].

Several analytical techniques can be used for the detection and characterization of nanoparticles. The most advanced techniques include scanning or transmission electron microscopes (SEM, TEM). They provide direct evidence of the size and shape of the nanoparticles, and in combination with coupled techniques, such as Energy-dispersive X-ray spectroscopy (EDX), information about the chemical composition can be obtained. Other optical-based techniques can also provide different physicochemical information of nanoparticles, although their usefulness depends on the specific technique and properties of the materials (composition, size, etc.). However, in most cases, these techniques require complex bench-top instrumentation leading to high-cost analyses, experiments taking long times which are only available to experienced users. Therefore, they are not very suitable if a rapid response is required, for example, when issues arise in industrial production or environmental/health risk situations.

Electrochemical techniques can address some of these issues since low-cost, user-friendly, portable potentiostats can be easily fabricated [11,12,13] and even self-powered devices are available [14]. Electrochemistry also provides good analytical sensitivity, which has proven convenient for environmental [15] or clinical analysis [16] and in combination with disposable single-drop electrodes become an excellent system for simple, cost-effective, and in situ analysis in decentralized settings. The main disadvantage is that information about the chemical composition cannot be readily obtained, although some selective measurements are possible by selecting an appropriate detection potential and hyphenated techniques, such as spectroelectrochemistry [17,18], which could provide chemical information. Electrochemical cells with conventional electrodes allow to make experiments in very controlled conditions, but they are not user-friendly and convenient for portable and in situ analysis. Screen- or stencil-printed electrodes are a great alternative to conventional systems for decentralized analytical applications [19]. They are usually fabricated within a planar solid substrate where the electrodes are printed using a conductive ink or paste passing by a specific mask with the appropriate shape. Ceramic substrates have been widely employed due to the possibility of ink curing at high temperatures, but novel substrates, such as plastic [20] or paper [21,22], with particular properties and lower fabrication costs have been substantially advanced in the last years.

Printed electrodes are also a great tool for the development of biosensing devices since they can be mass-fabricated at low cost and be used as single-use (disposable devices). This fact and the low sample volume required for a typical measurement are advantageous when dealing with biological samples to avoid cleaning electrodes from potential biohazards. These devices provide a convenient platform to carry out biological recognition processes in a simple way and enable the straightforward transduction of chemical events to direct electrical signals. Consequently, screen-printed and paper-based electrodes have been widely employed as transducers of (bio)sensing devices [23,24,25] for clinical [26,27], environmental [28], and food [29,30] applications. The elegant and attractive application of such screen-printed devices has also been demonstrated by fabricating body worn tattoo sensors for (bio)sensing in diverse fields of medical [31,32,33], security [34], and fitness [35]. In order to monitor a non-electroactive biological reaction involving the recognition of the analyte, an electrochemical signal related to that reaction should be provided. This can be achieved by using a label or tag, a chemical entity attached to the biological reaction that can be measured and provides quantitative information about the analyte [36]. Use of enzymatic labels is widespread due to their high sensitivity [37,38], but they require mediators and enzyme substrates, only work under specific biological conditions, and long-term storage could be an issue due to enzyme stability. The development of novel electrochemical labels to address these issues is a current scientific matter of interest [39,40], and nanotechnology could be a promising solution since many nanoparticles can provide electroactive signals, have high biocompatibility and stability, and can be easily functionalized [41]. Electrochemical detection of nanoparticles could also achieve high sensitivity and the versatility of having different materials, which allow multiplexing analysis, enabling the simultaneous detection of several analytes. Therefore, the electrochemical detection of nanoparticles plays a significant role in the new era of biosensing devices, and, particularly, in combination with disposable printed electrodes. It is worth mentioning that innovative detection methods of nanoparticles employed in most biosensing devices can also be potentially used in stand-alone applications for the detection of nanoparticles in regular or environmental samples.

In this review, the state of the art and possibilities of different strategies for the electrochemical detection and characterization of nanoparticles using printed electrodes (Figure 1) are described. Commercial or lab-made carbon screen-printed electrodes (SPEs) have been the most popular printed devices, but some paper-based electrodes have also been reported for the detection of nanoparticles. The electrochemical strategies described herein involve the use of these printed devices for the detection of nanoparticles from solutions, adsorbed on electrodes or attached to biochemical species after a bioaffinity reaction. However, the focus is placed on the electrochemical detection method itself and not on the final application. A previous review about strategies for electrochemical detection of nanoparticles in biosensing was published in 2008 [42]. The literature available for printed electrodes at that time was scarce, so most of the references dealt with conventional electrodes and a wider range of techniques. More recent reviews discussing electrochemical detection of some types of nanoparticles are available, but authors focused mainly on the application such as biosensing strategies or multiplexing analysis [39,43,44,45,46]. It is also worth to mention a general review on several analytical techniques for the detection and characterization of nanoparticles [9]. Readers are recommended to follow these review articles to increase their knowledge about those specific applications and complement the information addressed in this review.

## 2. Strategies for Electrochemical Detection of Nanoparticles Using Printed Devices

Different strategies for the electrochemical detection of nanoparticles have been organized in five different main groups (Figure 1): (a) direct detection methods, where the nanoparticles are directly detected by a redox process; (b) digestion methods, where the nanoparticle is dissolved to release their components to the solution, which are then detected; (c) methods where an electrocatalytic reaction is used as an indirect indicator for nanoparticle detection; (d) surface-enhanced methods, where the surface properties of the nanoparticles are exploited to enhance the detection; (e) nanoimpacts method, where individual nanoparticles are detected by stochastic collisions on the electrode surface. In the following sections, the main considerations of all these electrochemical strategies for the detection of nanoparticles using printed devices, which have been previously reported in the literature, are described.

### 2.1. Direct Detection Methods

Electrochemical detection of nanoparticles can be achieved by the direct oxidation or reduction of an electroactive component of the nanoparticle, typically a metal composing the nanoparticle. This direct oxidation or reduction could lead to the final electrochemical signal related to the concentration of nanoparticles for quantification purposes or as an initial stage to produce the preconcentration on the electrode surface, which would result in a more sensitive method for the subsequent signal recording (for example, in anodic stripping voltammetry (ASV) experiments). This technique presents advantages, such as avoiding the use of digestion or enhancing steps (*vide infra*) and requiring a simplified procedure since all the steps are performed in the same solution. In consequence, it is usually quicker than other non-direct methods. Since a direct electron transfer needs to be produced between the electrode surface and the nanoparticles, they have to be at a very close distance to enable electron tunneling [47,48,49]. This means that particles not close enough to the surface would not be able to be detected, making more difficult to get high sensitivity measurements. 

Gold nanoparticles (AuNPs) have been widely employed in different applications since they can be synthesized at various sizes, easily functionalized with biomaterials, present high biocompatibility and good stability. The direct electrochemical detection of AuNPs has been demonstrated and together with printed devices make a great platform as disposable (bio)analytical devices. Since these nanoparticles show great stability, their direct electrochemical detection usually involves a pre-oxidation step in order to generate the more electroactive Au(III) ions. This is only feasible at fairly high oxidation potentials in specific conditions, such as in the presence of hydrochloric acid solution, in order to facilitate the oxidation of Au(0) and formation of a chloride complex, AuCl_4_^−^. Then, the in-situ formed Au(III) species are electrochemically reduced again to Au(0) on the electrode surface (Figure 2), and a well-defined cathodic response is obtained using differential-pulse voltammetry (DPV), which is usually considered as the analytical signal related to the initial concentration of AuNPs. This method has been widely reported in many different biosensing systems using SPEs demonstrating its good performance [50,51,52,53]. Interestingly, this method has also been successfully applied inside the porous structure of paper-based carbon electrodes [54]. It is worth to mention that the AuNPs size has a strong influence on the electrochemical detection behavior [55]. It seems more challenging to achieve the efficient detection of small AuNPs when they are in solution due to the Brownian motions of small nanoparticles and the need of having the nanoparticles close to the electrode surface to enable the electron transfer. However, in biosensing applications, since the AuNPs are attached to the biological reaction, the Brownian motion is minimized, and they are usually in close contact to the electrode surface. In this case, smaller nanoparticles (5 nm) provide the best electrochemical response by the increased surface area compared to bigger particles. Since this method requires a high acidic solution (HCl) in order to enable efficient oxidation of the AuNPs, some approaches have been developed to avoid the use of this corrosive acidic media, which obviously would be a problem when used in decentralized settings. For instance, a NaNO_3_/NaCl solution, as a milder oxidizing agent in combination with AuNPs functionalized with PEG-based ligands to produce a less compact functionalization layer, has been proposed for the direct electrochemical detection of AuNPs with commercial carbon SPEs [56]. Results obtained were comparable to the typical strategy using HCl as detection media and AuNPs functionalized with citrate ligands. 

Silver nanoparticles (AgNPs) can be oxidized at lower potentials than AuNPs since it is a less noble metal, and a well-defined sharp voltammetric signal can be recorded. Some studies have shown that the oxidation signals for AgNPs are around 100 times higher than for AuNPs of same diameter and concentration [57]. Furthermore, Au(III) species are only stable under some specific conditions; meanwhile, Ag(I) could be present as soluble species forming a complex or as the insoluble precipitate. The electrochemical response depends highly on the electrolyte medium [58,59], but this also offers more versatile experimental conditions where the AgNPs can be electrochemically oxidized by the direct method and could open the possibility to develop different tailored strategies for specific applications. As an opposite effect, AgNPs are usually less stable than AuNPs, are prone to aggregate in several media and, therefore, their use in applications, such as biosensing, is more challenging. Two main strategies using printed electrochemical devices have been reported for the direct electrochemical detection and characterization of AgNPs. Firstly, the direct stripping of silver from the AgNPs using voltammetric methods is the simpler one. However, to get a sensitive enough signal, AgNPs need to be efficiently close to the electrode surface, in a method usually called voltammetry of immobilized particles. This has been exploited for the quantification and characterization of AgNPs from commercial products using SPEs [60]. Interestingly, the oxidation peak potential recorded from linear sweep voltammetry (LSV) could also be related to the diameter of the nanoparticles. For this, screen-printed pseudoreference electrodes need to be modified with an AgCl layer and Nafion^®^ in order to get a stable reading of the electrode potential. A disposable sticky electrode was also reported for the detection of AgNPs in seawater [61]. Carbon SPEs were modified with cysteine by electrodeposition to exploit the strong interaction between the Ag-S atoms in order to immobilize the AgNPs close to the electrode surface allowing the sensitive electrochemical detection by direct LSV. This method proved useful for monitoring AgNPs directly in seawater. The second strategy involves the initial pre-oxidation of the AgNPs by applying a positive potential, and, then, ASV with DPV detection is employed following an initial deposition step to reduce the Ag(I) atoms formed in the pre-oxidation step. This method leads to the preconcentration of the metal component on the electrode surface and is usually more sensitive than the direct stripping method. It has been successfully employed in biosensing devices using SPEs [62,63]. An interesting approach was reported using a dual screen-printed working electrode device [64]. Working electrodes with different size were employed, and the biological reaction was carried out using both electrodes as heterogeneous surfaces. AgNPs, which were used as detection label, were electrochemically oxidized to silver ions by applying a positive potential in both electrodes. Then, only the smaller electrode was used to preconcentrate the silver by electrochemical deposition, and the detection was carried out by voltammetric stripping. This strategy enables the use of a large surface area for the biological reaction, but the use of a small electrode for the detection of nanoparticles allows to enhance the faradaic-to-capacitive current ratio (signal/noise) leading to an improvement in the limit of detection by 9X. An alternative method for the electrochemical detection of AgNPs is by exploiting the galvanic exchange process [65], as illustrated in Figure 3. In this case, a paper-based device with carbon electrodes was modified with a small amount of gold. When AgNPs are close to the gold within the electrode, a galvanic exchange occurs between Au and Ag: a fraction of Au(0) is oxidized to Au(III), and the latter can react with the AgNPs to form Ag(I), which would be more easily detected by LSV-ASV. This method avoids the use of an added oxidant, which has been usually employed as a precedent step in assays using paper-based devices [66] (*vide infra*).

AuAg alloy nanoparticles present the best of both worlds in relation to AuNPs and AgNPs since they provide not only the possibility of an easy and sensitive detection by the silver oxidation signals but also present better stability than typical single-component AgNPs. These alloyed nanoparticles can be characterized and detected by a direct electrochemical method as demonstrated using polyester SPEs [67]. The detection method is performed in a chloride-containing solution (PBS, pH 7.4), which is able to make controlled corrosion (Cl^−^ and O_2_ are essential) of the silver component (Figure 4), and detection is carried out by deposition-stripping steps. The presence of gold also helps to preconcentrate silver in the deposition step since an underpotential deposition (UPD) process is enabled. This leads to anodic oxidation signals one order of magnitude higher for silver in AuAg particles than those obtained for AgNPs. Therefore, they are a good alternative to be used as an electrochemical label in biosensing [68].

Quantum dots (QDs) are crystalline nanoparticles usually formed by different metallic semiconductor components, which are widely used because of their special optical properties. The most employed are based on Cd, Pb or Zn metals, which all have a good electroactivity, and, therefore, can be electrochemically detected. These nanocrystals are usually detected by using an initial digestion step to release the metal ions to the solution (*vide infra*), but they can also be directly detected by electrochemical methods. This is possible because the surface of QDs could have some more available metallic atoms due to structural defects that can be directly reduced and preconcentrated on the electrode surface. Then, the stripping signal from the preconcentrated metal can be related to the concentration of nanoparticles. However, this kind of surface atoms are in the minority in the whole structure and depend on the surface defects and other structural characteristics. Thus, the amount of metal able to be reduced this way is not very high, and the sensitivity of these methods is usually lower than using a digestion step. Regardless, QDs direct electrochemical detection has been shown to be useful for different applications using printed electrodes as detectors. For instance, commercial SPEs were used for the direct detection of CdS QDs in a neutral solution (pH 7) by ASV and square-wave voltammetry (SWV) with an initial direct reduction of Cd(II) from the nanoparticle [69]; and successfully applied in biosensing [70]. The direct detection of CdS QDS was also demonstrated in acetate buffers (pH 4–5) with applications ranging from microfluidic devices with integrated screen-printed electrodes [71] to immunosensing [72]. Acetate is the most employed electrolyte for electrochemical detection of heavy metals, and it is also widely employed for QDs detection. Even if these direct QDs detection methods are not using a specific digestion step, it may be possible that the QDs are not very stable under this slightly acidic media, and more metallic ions could be released to the solution than using more neutral or alkaline pH, where these nanocrystals have usually higher stability. Direct detection of QDs is not only possible with nanoparticles of a single component, and core-shell CdSe@ZnS QDs have also been detected using polyester and paper SPEs in a solution at neutral pH [54]. Interestingly, paper electrodes showed similar analytical performance for nanoparticles detection than polyester SPEs, but with the inherent advantages of paper devices.

Nanoparticles can also work as carriers of electroactive species and be quantified by recording an electrochemical signal coming from the loaded species. These nanoparticles could be loaded with species of different chemical nature, enabling their utilization in multiplexing biosensing since they would produce two different electrochemical signals compared to single metal nanoparticles, such as AuNPs or AgNPs. Some examples have been reported where printed electrodes have been used for the direct detection of electroactive-species loaded nanoparticles. For instance, paper-based analytical devices coupled with SPEs were employed for the detection of nanoporous gold nanoparticles loaded with thionine [73]. Two assay zones were generated on the paper device by wax printing, and to carry out the detection, they were put in contact with the printed carbon electrode by folding the paper. The use of nanoporous particles clearly helped to create high surface-area carriers enhancing the loading and creating highly sensitive detection labels. These nanoparticles were detected by the electrochemical signal produced by thionine recorded with the DPV technique. Metal-ion loaded nanoparticles are also a good alternative since the possibility to detect metals by electrochemical methods is very powerful. ASV is a very sensitive technique, and different metals produce stripping signals at different potentials, enabling their selective identification and quantification. AuNPs modified with cysteine were loaded with two different metal ions (Ag^+^ and Pb^2+^) and were successfully detected by a direct method with foldable paper-based electrodes [74]. Titanium phosphate nanoparticles (TiPNPs) are also a great option to introduce metals into their structure and be used as very efficient electrochemical labels in biosensing. These nanoparticles have a porous structure with phosphate groups on the surface, which can be involved in a cation-exchange reaction to attach the metals into the nanoparticles [75]. Then, these nanoparticles, initially non-electroactive, can be directly detected and quantified by square-wave ASV using commercial SPEs [76] and be applied as labels in biosensing [77]. Furthermore, tailored nanoparticles can be obtained by enhancing the cation-exchange reaction during the synthesis leading to nanoparticles with a high-loading of metal ions. This enhanced synthesis in combination to inducing a more efficient metal extraction during the electrochemical detection by reversing the cation-exchange reaction makes the electrochemical detection of these nanoparticles highly sensitive [78]. Different metals can be introduced, such as Cd, Ag, Hg, Cu, Bi to create tailored nanoparticles for specific applications with enhanced sensitivity and selectivity for use in multiplexing biosensing [78] since the different loaded metals can be simultaneously detected (Figure 5).

### 2.2. Detection with a Preceding Digestion Step

An alternative option usually more sensitive than the direct method involves releasing the metallic atoms from the nanoparticle structure to the solution since metallic nanoparticles are composed by a relatively large number of atoms that are usually electroactive at appropriate potential ranges. This approach avoids the requirement from the direct method of close contact between the nanoparticles and electrode surface to enable the electron transfer since the atoms can be easily transported to the electrode surface by diffusion or convection. Acidic digestion is usually the most employed strategy to cause the digestion of the nanoparticles and releasing of metallic ions, but the use of a strong oxidant has also been reported for some materials. Then, the released metal can be electrochemically detected usually by ASV involving an initial electrodeposition step for preconcentration onto the electrode surface. The potentiality of this method relies on the chemical agent employed to cause the nanoparticle digestion as quickly and with the highest yield as possible. In this regard, the digestion of AuNPs has not been recently reported as a method for the electrochemical detection of these nanoparticles using printed devices since it is only possible with good efficiency in very harsh conditions such as using Br_2_/HBr solutions [79]. AgNPs and QDs (formed by chalcogenides of Cd, Pb, Zn or other metals) are the nanoparticles with more possibilities to be detected by this method and have been specially employed as electrochemical labels for biosensing purposes. 

Since the oxidation of AgNPs could happen under milder conditions than for AuNPs, several strategies to generate Ag(I) ions from AgNPs in order to improve the electrochemical detection in screen-printed devices have been reported [80,81]. For instance, a chemical agent such as SCN^−^ is able to facilitate the chemical oxidation of AgNPs to generate AgSCN due to the high complexing power of this anion. However, this method involves the incubation of the AgNPs with the complexing ligand for one hour to provide sensitive detection. Furthermore, after the incubation step, an aliquot of the resulting solution is transferred to the SPE to carry out the detection by ASV, and, therefore, the procedure entails several steps making its use not very relevant for a simple, quick point-of-care analysis system. Anyway, this strategy has been employed to monitor different immunoassays [80,81].

Interestingly, the versatility of paper-based analytical devices (PADs) is apparent in the development of novel enhanced electrochemical methods for nanoparticles detection. For instance, an oSlip PAD (Figure 6) was developed consisting of several paper layers to carry out in subsequent steps: a biological reaction, chemical oxidation of AgNPs, and electrochemical detection [66]. A chemical oxidant, such as KMnO_4_, can be placed in a specific moveable paper layer that is easily put on contact with the surface where the reaction was carried out in order to oxidize the AgNPs at a specific time and location by simply slipping the paper layer. Detection is performed in a chloride solution, but the precipitation of AgCl does not seem to occur in the presence of the MnO_4_^−^ as it increases the solubility. Then, the Ag^+^ released from the nanoparticles is detected by ASV. This device has been successfully applied in different biosensing systems [66,82]. An improved strategy with a similar paper-based device but avoiding the use of KMnO_4_ as oxidant was also reported [83]. KMnO_4_ is a very efficient agent for the oxidation of AgNPs, but it is also a very strong oxidant able to generate O_2_ from water oxidation forming MnO_2_ as a subproduct of the reaction. This compound acts as a catalyst for the decomposition of MnO_4_^−^, and, therefore, the solution with the oxidant agent is not very stable before use. Besides, MnO_2_ is insoluble and may form an insulated layer on the electrode surface. Interestingly, using a milder oxidation agent, such as ClO^−^, is possible to avoid the negative effects mentioned for KMnO_4_, but, at the same time, still producing the oxidation of AgNPs in an effective way.

Since QDs can be formed by different metals, they have been widely employed as a label for multiplexing assays in order to detect simultaneously several analytes. The great performance of the ASV electrochemical method for heavy metals also provides very sensitive detection of these nanoparticles. A conventional and widely reported strategy for QDs detection in biosensing involves carrying out the biological reaction in a reaction container and then an acidic solution is added to produce the nanoparticles dissolution releasing the metallic cations to the solution. Then, an aliquot of this solution is transferred to a conventional electrochemical cell for the detection step [84,85]. It is evident that this methodology is inadequate for decentralized settings, where the aim is providing a simple, low-cost, point-of-care analysis with low reagent consumption. Nevertheless, some works where SPEs were employed as the detection platform instead of a conventional electrochemical cell have been reported [86]. Other, more complex strategies have also been used, such as cutting the test zone of an immunochromatographic test, placing it in HCl solution to induce the release of metals, and then the solution is transferred to a commercial SPE for the detection [87]. This is a complex and time-consuming process losing all the advantages that both the immunochromatographic strips or SPEs provide. In contrast, an interesting strategy in order to simplify the electrochemical detection of QDs with a digestion step in disposable carbon SPEs (Figure 7) was developed. This strategy allowed to perform the biological reaction, the acid digestion of QDs, and the electrochemical detection using the same SPE [88]. Thus, it is possible to avoid the use of external reaction containers to carry out the biological reaction and the subsequent transfer of solution after the digestion step. This method simplifies the overall process and facilitates their use towards a point-of-care system. Bismuth ions are also added to the detection solution to increase the sensitivity and reproducibility of the ASV of the metals. It was found that the acidic digestion was essential to achieve the detection of QDs with high sensitivity, and a 15X increment of the electrochemical signal was observed compared to the direct detection [88]. This method has been successfully applied for the detection purposes in immunosensing [89,90]. Interestingly, the characterization of QDs to get information, such as the concentration in solution or the size, can also be achieved following a similar method. Size results obtained from the electrochemical detection were comparable to the standard optical method [91]. Embedding the bismuth precursor on the screen-printed electrode surface has also been demonstrated as a good strategy to avoid the ex situ addition to the electrolyte solution, and it has been successfully employed in different biosensing devices using polyester carbon SPEs [92,93]. Similar methods for electrochemical detection of QDs using carbon SPEs integrated within the channel of microfluidic devices [94,95] or using paper-based or disposable devices [96,97] have also been proposed.

The electrochemical detection of QDs can be further enhanced exploiting the magnetohydrodynamic (MHD) effect when applying a magnetic field to the electrode during the electrodeposition of the metals [98]. This is achieved by placing two magnets in parallel to the electrode surface of SPEs (Figure 8). The MHD effect induces convection to the solution which is directly proportional to the generated electrochemical current and the magnetic field. Since the aim is to detect low analyte concentrations, Fe(III) as an inert species is added to the solution at high concentration. During the electrodeposition of Cd released from the QDs, Fe(III) is simultaneously reduced to soluble Fe(II) generating a high current that induces a strong MHD in the presence of the magnetic field. This effect increases the mass transfer of Cd(II) to the electrode surface, enhancing the deposition efficiency and leading to a more sensitive detection with a limit of detection 2X lower than the conventional detection strategy described above. This is interesting since planar printed devices are usually employed using a static drop of the sample, and in this configuration, external convection cannot be applied. This method was successfully applied for the detection of QDs in aqueous solutions and for biosensing purposes. 

### 2.3. Electrocatalytic Detection Methods

Electrocatalysis is essential in many relevant processes, especially in energy conversion systems involving reactions, such as the hydrogen evolution reaction (HER), the oxygen reduction reaction (ORR) or the oxygen evolution reaction (OER), among others. Many nanomaterials have been reported as excellent materials to enhance electrocatalytic processes by facilitating the electron transfer [99]. This can be exploited in several ways and open the possibility to selectively discriminate catalytic from non-catalytic materials. Furthermore, the electrocatalytic behavior of nanoparticles could also depend on composition, shape, and size [100] since a high surface area and efficient active sites play a significant role on the final response. The possibility to differentiate a non-catalytic electrode surface from electrocatalytic nanoparticles can be employed for the quantification of those nanoparticles. More interestingly, it opens the possibility to detect nanoparticles with a very noble character, such as Pt, Pd or Ir, that can be challenging to detect by conventional methods (direct or digestion methods). This indirect method has been applied using low-cost printed electrochemical devices and different electrocatalytic reactions. Printed electrodes are usually fabricated with carbon materials, which are not electrocatalytically active for many electron transfer reactions. Therefore, the electrocatalytic reaction could be selectively detected at lower potentials in the presence of the active nanoparticles [29]. The main advantage of this method is avoiding the digestion step by acidic/oxidant solutions, and, therefore, the detection can be carried out directly with good sensitivity. 

The most employed electrocatalytic reaction for indirect nanoparticle detection with printed devices has been the HER. Several metallic nanoparticles can facilitate this reaction, but AuNPs have been widely employed. This is probably because of the high biocompatibility, stability, and the ability to biofunctionalized these particles compared to other materials. The detection method is usually carried out by chronoamperometry in high concentration acidic solution (around 1 M). Then, a reduction potential is applied for a specific time until reaching a quasi-stationary reduction current, which is related to the amount of AuNPs present. This method was compared to the direct electrochemical oxidation/reduction using paper-based electrodes achieving a slightly better limit of detection with the electrocatalytic method for quantification purposes [54]. This method has been applied in biosensing applications [101,102,103,104] using carbon SPEs fabricated on polyester. Figure 9 shows a scheme of the detection of AuNPs by the electrocatalytic HER using chronoamperometry. Silver-modified TiPNPs are also possible to quantify with a similar method using the HER as an indicator reaction [76]. In this case, although the nanoparticles are loaded with silver ions, after applying a negative potential, AgNPs are formed in situ on the electrode surface, which electrocatalyze the HER. A slightly better limit of detection was found for a direct voltammetric method, probably due to the usual great performance of the preconcentration and stripping of silver using a voltammetric detection.

Other electrocatalytic reactions have also been exploited for nanoparticle quantification at a lesser extent. AgNPs are also able to catalyze the reduction of H_2_O_2_, and this reaction was employed for nanoparticles detection [105]. In this case, an origami paper device, modified with gold nanorods as electrodes, was employed for the detection of porous zinc oxide nanospheres loaded with AgNPs by chronoamperometry but at a lower potential than for the HER, which makes this process a bit more selective (since less possible impurities can be reduced at lower potentials). Oxygen-involving reactions, such as the ORR and OER, have also been employed for the electrocatalytic detection of nanoparticles using SPEs. For instance, IrO_2_ nanoparticles were detected by the electrocatalytic OER [106]. This material is known to be one of the most active materials for this reaction in acidic media [107], and their detection was also possible in neutral pH, which avoids the strong acidic media needed for the electrocatalytic detection of AuNPs by the HER. The ORR can also be employed for the electrocatalytic detection of palladium nanoparticles in neutral media and using SPEs [108]. In this case, a voltammetric detection was carried out since the inherent solubility of O_2_ in aqueous solution makes possible the recording of a diffusion-controlled voltammetric peak as the analytical signal. In another interesting concept, platinum nanoparticles (PtNPs) were detected by means of the catalytic oxidation of hydrazine [109]. Interestingly, the detection was carried out at open circuit potential (OCP) using SPEs. Hydrazine was oxidized in the presence of the PtNPs, and since the initial conditions change (N_2_H_4_ is depleted, and N_2_ and H^+^ are generated), a change in the OCP is observed. 

### 2.4. Surface-Enhanced Detection Methods

Surface-enhanced methods have also been employed for the detection of nanoparticles leading to the recording of amplified signals compared to the direct detection. These methods usually involve the catalytic reduction (and deposition) of metal on the nanoparticle surface and the posterior detection of the formed deposit. This strategy can be carried out by a chemical reduction utilizing a specific solution called enhancer [110] or by electrodeposition [111]. The new metallic deposit grown on the nanoparticle surface is usually bigger than the initial nanoparticle and composed of a material that can be detected more efficiently, leading to more sensitive detection. This process has been reported for silver or copper deposition on gold nanoparticles [111,112,113], although only a few examples have been reported using printed electrochemical devices as the detection platform. This fact is probably due to the challenging control of the selective deposition of the new metal and the need to avoid the deposition on the non-selective electrode surface. Electrochemical printed devices typically use solid-state pseudoreference electrodes, and, therefore, the working electrode potential may be difficult to control with high precision during the required selective electrodeposition step. 

Some works have been reported using surface-enhanced methods for nanoparticles detection with printed electrochemical devices. Using a silver enhancing solution in order to induce the silver deposition on AuNPs with the subsequent detection by silver stripping was employed as detection strategy [114]. An enzymatic reaction leading to silver reduction is also an alternative approach to produce the deposition of AgNPs on the surface of AuNPs [115], which were quantified using SPEs by recording the silver stripping signal. Another work, also for the detection of AuNPs, was reported following a multistep amplification strategy [116]. Sub-10 nm AuNPs were enhanced acting as nucleation seeds for chemical deposition of a gold shell (using HAuCl_4_ and ascorbic acid as reducing agent). After this growth process, a solution containing silver and gold salts in the presence of NH_2_OH was used for increasing the size and leading to high-surface-area spiky nanoparticles. Finally, a silver enhancement step was carried out with AgNO_3_ and hydroquinone, and the deposited silver was electrochemically detected by stripping. Obviously, this methodology leads to enhanced nanoparticles with a large amount of silver deposited on the surface and to very sensitive detection. However, it entails multistep chemical reactions taking several minutes, making this detection strategy very difficult to apply in the point-of-care analysis, which is one of the main applications using electrochemical printed devices. 

Quantum dots have also some particular surface properties, and their very small size (typically between 3–5 nm) make them very appropriate for surface enhancing detection methods. A novel electrochemical method using commercial SPEs for the quantification of CdSe/ZnS QDs by the selective electrodeposition of silver on the QDs surface was reported [117]. Selecting an appropriate potential and experimental conditions, silver can be selectively electrodeposited only on the QDs surface (Figure 10A), avoiding the deposition on the carbon electrode due to the strong interaction between silver and QDs. A selective process is also observed for the silver electrochemical stripping since it occurred at a higher potential than the silver stripping from carbon (Figure 10B), making possible the differentiation of both processes. This is in contrast to AuNPs where the selective stripping is not possible since the potential is similar when the silver is stripped from the carbon electrode surface or the AuNPs. Furthermore, this method enables the direct detection of QDs avoiding the acidic digestion step, which was a requirement to achieve a high sensitivity (*vide supra*). Silver-enhancement of QDs results in a limit of detection of one order of magnitude lower than the acidic digestion method typically employed for QDs detection.

A very interesting electrochemical behavior of copper species in the presence of QDs was recently reported [118], which can be exploited for the quantification of the nanoparticles [119] using screen-printed carbon electrodes (SPCEs). Cu(II) reduction in ammonia media at carbon electrodes can electrogenerate Cu(I) species but they can be easily oxidized in the presence of O_2_ resulting again in the formation of Cu(II), and the backward oxidation of Cu(I) to Cu(II) is not observed in a voltammogram (Figure 11A). Interestingly, Cu(I) species are stabilized in the presence of CdSe/ZnS QDs, and a significant oxidation current can be recorded. The amount of Cu(I) and, therefore, the magnitude of the oxidation current, is directly proportional to the concentration of QDs (Figure 11B). A difference with the silver-enhancing method is that, in this case, there is no electrodeposition since Cu(I) is a soluble species under these conditions. This is a very innovative and promising method for QDs quantification since it is possible to achieve the direct electrochemical detection of these nanoparticles avoiding the acidic digestion step and leading to exceptional sensitivity, with a limit of detection even lower than the silver enhancing method.

### 2.5. Detection by Nanoimpacts

Electrochemical detection of nanoparticles by the nanoimpact method [120,121] is an emergent strategy for the characterization and quantification of individual nanoparticles and even enables the detection of single entities of biochemical species [122]. This method entails the stochastic detection of particles colliding against an electrode surface, providing information about individual entities instead of average ensembles of particles, such as in the conventional methods previously described. The detection can be carried out by different strategies (Figure 12): direct oxidation/reduction processes of the nanoparticle components (usually called particle coulometry) [123] or indirectly by using an electrocatalytic reaction [124] or detecting the blockage [125] of the electrode surface after the particle impact. Interestingly, the obtained signals can provide relevant information about each individual nanoparticle colliding with the electrode surface, such as the size, surface charge or transferred electrons, and the frequency of collisions is usually related to the concentration of particles in the sample [120]. This method allows obtaining in situ information about the properties of single nanoparticles in a very simple way and at a much lower cost than other methods, such as electron microscopy. Micro-or nanoelectrodes are usually employed to detect the nanoimpacts since they provide a higher signal/noise ratio due to the low capacitive currents. However, the detection of nanoparticle impacts has also been reported using carbon SPEs, enabling this simple platform to be used in frontier nanoscale electrochemistry. A significant difference is that SPEs can be used in a horizontal configuration, allowing the particles to collide with the electrode surface not only by diffusion (thermal Brownian motion) but also by gravitational forces. 

Characterization and quantification of AgNPs are the most reported applications of the nanoimpact method since AgNPs can be easily oxidized at an appropriate anodic potential. SPEs can also be potentially employed for the characterization of AgNPs by this method. A study with AgNPs of sizes between 10 and 107 nm was carried out with interesting conclusions [126]. It was proposed that agglomerates of several nanoparticles impact the electrode surface at the same time, which is expected in a chloride solution, and not all the particles within the agglomerates were completely oxidized. The proposed number of particles per impact was indirectly proportional to the nanoparticle size. There is one main limitation for using SPEs since the detection of nanoparticles of low size is challenging due to the high background charge. Nanoparticles from other materials, such as metal sulfides or molybdenum, can also be detected by similar methodologies. Colloidal particles of PbS and CoS were detected by the nanoimpact method using cathodic particle coulometry by applying a constant potential with an ability to reduce the metal from the particle when the particle collides with the surface [127]. The size determined by the electrochemical method was in good agreement with the size determined by electron microscopy imaging. Molybdenum nanoparticles were also detected and counted by analyzing the current spikes caused by the oxidation to molybdenum oxide [128]. Authors propose a 6 e^−^ oxidation to MoO_4_^2−^, with partial oxidation already happening at a low potential (0 V vs. Ag/AgCl) and complete oxidation at higher potentials (+0.5 V). Size determination was also well correlated to data obtained from electron microscopy, demonstrating once again the usefulness of this method. 

Nanoparticles with electrocatalytic properties can also be detected by nanoimpacts in an indirect way using the electron transfer involved in the electrocatalytic reaction instead of a direct oxidation/reduction of the particle. This is possible when the electrode surface by itself is not able to produce the electrocatalytic reaction. A low-activity material, such as carbon, is usually employed, and when a catalytic particle collides with the electrode surface, the electrocatalytic reaction is enabled. Therefore, even if the nanoparticle by itself is not oxidized/reduced, a current spike that is related to the electrocatalytic reaction within the particle is recorded. In the same way, for the macroscale electrocatalytic detection of nanoparticles, the HER has been widely employed. Several materials, such as black phosphorus (BP) nanoparticles [129] and layered transition metal dichalcogenides (TMDs), [130] have been detected and quantified using the HER as an indicator reaction. These materials show high activities towards the HER, sometimes close to the most active noble metals. Interestingly, the BP was employed as an electrochemical label in a bioassay since the frequency of nanoparticle impact can be related to the amount of BP and then to the analyte recognized by the biochemical reaction. Meanwhile, the TMDs could also be detected by anodic particle coulometry since their oxidation is possible from M^4+^ to M^6+^, providing a double method for their detection (Figure 13). A similar collision frequency was found for both methods with these particles, which suggests that the same particles able to catalyze the HER are also electroactive. The ORR has also been used for the detection of nanoparticles by nanoimpacts [131]. In this case, a composite material formed by graphene sheets modified with Pt/Fe or Fe nanoparticles was employed. Therefore, graphene sheets were possible to be detected by this modification with metal nanoparticles, opening the door for the detection of graphene sheets by nanoimpacts. Since current spikes were recorded instead of stairs, it suggests that the graphene nanosheets do not remain on the electrode surface after the impact and return to the solution, which is interesting to avoid the change of the electrode surface nature.

## 3. Conclusions and Outlook

As a consequence of the fast growth rate of nanotechnology in different scientific and technological scopes in the last years, it is clear that the analytical detection and characterization of nanoparticles are essential for different purposes. They entail a possible environmental risk due to their increased use in commercial products but also provide excellent features in applications, such as point-of-care biosensing. Therefore, readily available, rapid, and cost-effective analytical methods for the detection of nanoparticles are urgently needed. A promising approach to cost-effective, user-friendly and portable detection of nanoparticles can be achieved by marrying the inherent attributes of electrochemistry with those of disposable printed devices. 

The main aim of this review has been to highlight the new developments in electrochemical strategies that enable the quick and sensitive detection of nanoparticles of different chemical nature using printed electrodes. Direct detection of electroactive nanoparticles provides a rapid and easy method that can be useful in applications where the sensitivity is not critical. For those cases, preceding digestion of the nanoparticle to release electroactive species to the solution may be needed, but it also increases the complexity of the procedure. Surface-enhanced methods provide great sensitivity avoiding the digestion step, but they are only possible with nanoparticles with particular surface properties. Another alternative is the use of electrocatalytic reactions as an indicator facilitated by the nanoparticles. This method enables the sensitive detection of nanoparticles that may not be directly electroactive in an appropriate potential range. One of the approaches that have been widely exploited in the last years is the nanoimpact method, which provides information about the concentration and properties of nanoparticles from a single entity approach. In conclusion, all these electrochemical strategies for the detection and characterization of nanoparticles cover a wide range of analytical possibilities that can be used for different applications. 

Electrochemical detection of nanoparticles is a very promising and helpful technique, but some challenges remain ahead in order to make this technique as universal and useful as possible. For instance, many of the direct, digestion or electrocatalytic-based methods require solutions with strong acidic or oxidant character. Replacing these solutions with greener and milder alternatives that can be active in near neutral pH could be of great importance, especially in clinical or in situ settings. The detection of non-electroactive nanoparticles is also a challenge to make electrochemical detection a universal method. It is worth to mention that the nanoimpact method has been successfully employed for the detection of non-electroactive particles [125,132]. This is possible with non-conducting nanoparticles, which, after colliding with the surface, block the electron transfer between the electrode and the electroactive species in the solution. Developing similar strategies for a wide range of non-electroactive nanoparticles could make electrochemical methods more universal. The nanoimpact method provides rapid information of individual nanoparticles and, therefore, shows a better picture of the heterogeneities of the sample. This is only possible with very complex and high-cost techniques, such as electron microscopy, and it is easy to anticipate the great relevance that the nanoimpact method will have in the coming years. Regarding electrode devices, paper-based substrates are the new star in this field because of the versatile and low-cost fabrication methods. Although promising, there is still a need to demonstrate that the paper-based devices can be employed for the same successful strategies employed in solid hard surfaces with the same efficiency and sensitivity. Of course, exploiting the microfluidic, porous characteristics of these devices could also lead to the development of innovative detection methods from different nature of the established ones in conventional devices. 

The coming years look very promising for the development of novel advanced approaches dealing with the electrochemical detection of nanoparticles using printed devices. Enhanced strategies coupled with the excellent inherent properties of electrochemistry in printed devices could evolve this rapid and cost-effective analytical tool into the favorite initial screening method of various experimentalists.

## Figures and Tables

**Figure 1 biosensors-09-00047-f001:**
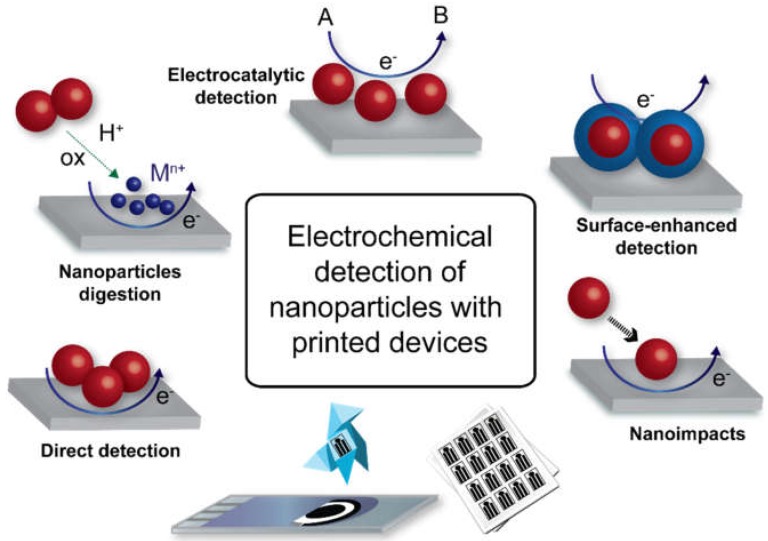
Scheme illustrating the different electrochemical strategies described in this review for the detection of nanoparticles using printed devices.

**Figure 2 biosensors-09-00047-f002:**
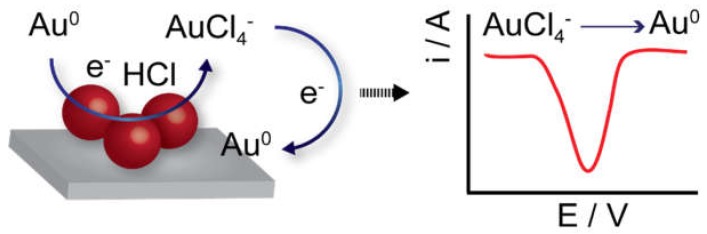
Scheme illustrating the direct electrochemical detection of gold nanoparticle (AuNP) by initial oxidation in HCl to form AuCl_4_^−^ and subsequent reduction again to Au(0). This reduction leads to a voltammetric cathodic signal that is proportional to the concentration of AuNPs.

**Figure 3 biosensors-09-00047-f003:**
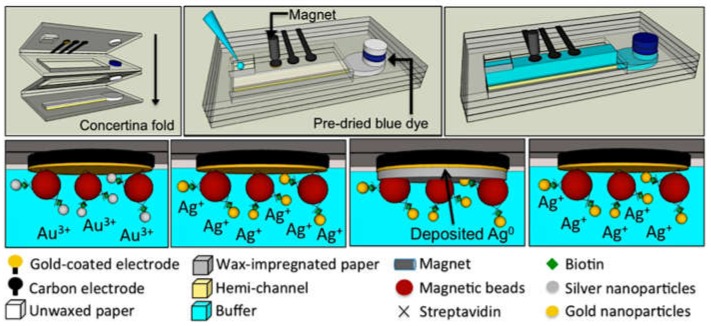
Scheme of the paper-based device used for the detection of silver nanoparticles and the galvanic exchange reaction that facilitates the electrochemical detection of the nanoparticles in the presence of gold-plated electrodes. Adapted with permission from [65] Copyright 2016 American Chemical Society.

**Figure 4 biosensors-09-00047-f004:**
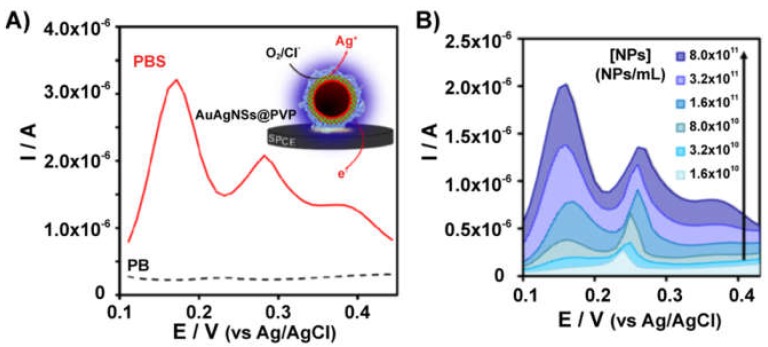
(**A**) Scheme of the detection of AuAg alloy nanoparticles (NPs) and voltammetric response showing the effect of chloride to achieve a sensitive detection signal with the carbon screen-printed electrode (SPE). (**B**) Voltammetric profiles for increasing concentrations of AuAg nanoparticles. Adapted with permission from [67] Copyright 2018 American Chemical Society—Open Access ACS AuthorChoice.

**Figure 5 biosensors-09-00047-f005:**
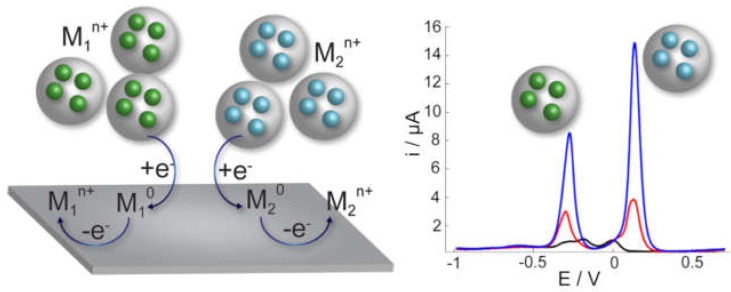
Scheme of the simultaneous electrochemical detection strategy for nanoporous titanium phosphate nanoparticles loaded with different metals and corresponding anodic stripping voltammograms. Adapted with permission from [78] Copyright 2017 Elsevier.

**Figure 6 biosensors-09-00047-f006:**
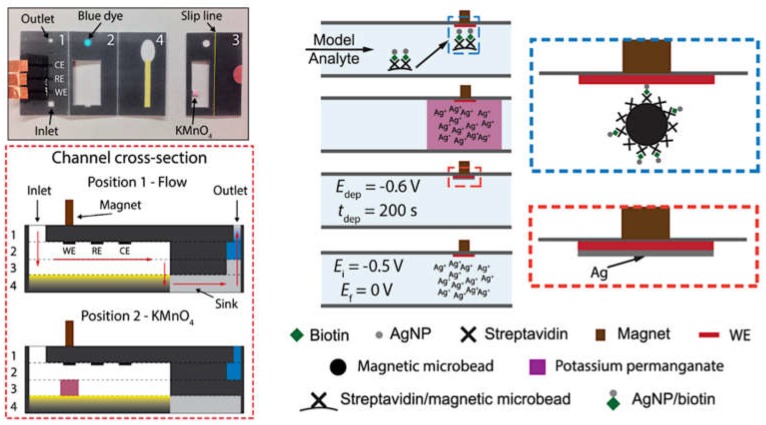
Image of the oSlip paper analytical device, scheme of the device cross-section when the oxidant layer is placed close to the working electrode (WE), and scheme of the electrochemical strategy for the detection of silver nanoparticles (AgNPs) with this device. Adapted with permission from [66] Copyright 2014 American Chemical Society.

**Figure 7 biosensors-09-00047-f007:**
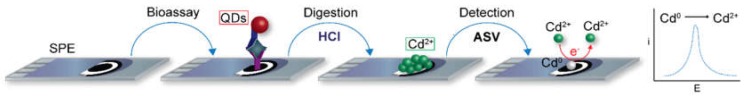
Schematics of the in-situ detection of quantum dots (QDs) used as a label for electrochemical biosensors where the biological reaction, digestion step, and electrochemical detection by anodic stripping voltammetry (ASV) are all carried out using the same screen-printed electrode. SPE: screen-printed electrodes.

**Figure 8 biosensors-09-00047-f008:**
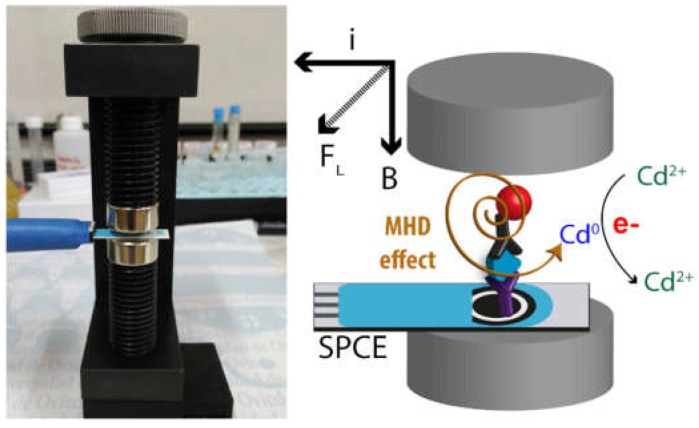
Picture of the magneto-electrochemical support to apply a magnetic field to screen-printed carbon electrodes (SPCEs) and schematic of the enhanced electrochemical detection of quantum dots (QDs) using the magnetohydrodynamic (MHD) effect for Cd electrodeposition and stripping. Reprinted with permission from [98] Copyright 2017 The Royal Society of Chemistry.

**Figure 9 biosensors-09-00047-f009:**
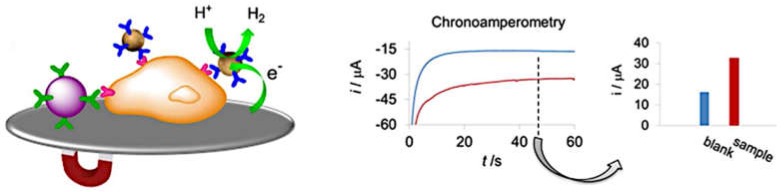
Scheme of the electrochemical detection strategy of gold nanoparticles (AuNPs) using the hydrogen evolution reaction electrocatalyzed by the nanoparticles with the signal recorded by chronoamperometry. Adapted with permission from [102] Copyright 2012 American Chemical Society.

**Figure 10 biosensors-09-00047-f010:**
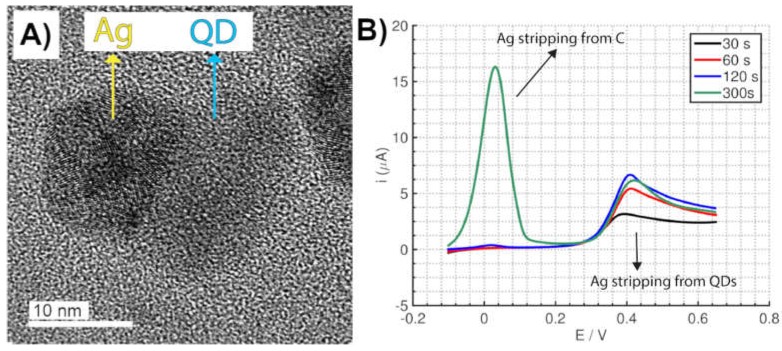
(**A**) Transmission electron microscopy image of silver electrodeposited on the surface of a quantum dot (QD) nanoparticle. (**B**) Selective voltammetric stripping of silver from QDs in comparison to stripping from carbon electrode. Electrodeposition at short times leads to selective deposition of silver on QDs. Adapted with permission from [117] Copyright 2016 American Chemical Society.

**Figure 11 biosensors-09-00047-f011:**
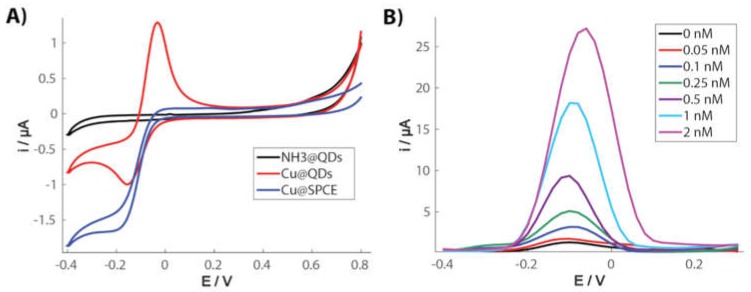
(**A**) Cyclic voltammetry of initial Cu(II) illustrating the influence of quantum dots (QDs) to stabilize the electrogenerated Cu(I) since the reoxidation signal to Cu(II) is observed compared to the electrode without nanoparticles. (**B**) Voltammetric signal of Cu(I) oxidation in the presence of increasing concentrations of QDs. Adapted with permission from [119] Copyright 2017 Elsevier.

**Figure 12 biosensors-09-00047-f012:**
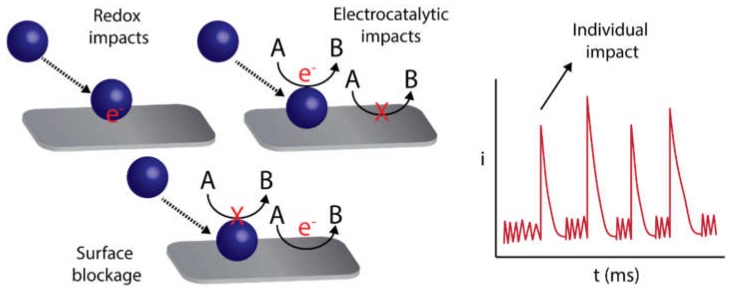
Different electrochemical strategies to detect collisions of individual nanoparticles and the typical stochastic amperometric response recorded when nanoparticles collide with the electrode. Adapted with permission (CC BY 4.0 license) from [122] MDPI.

**Figure 13 biosensors-09-00047-f013:**
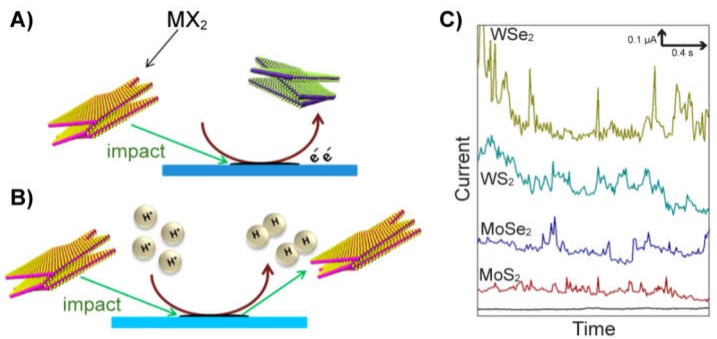
Scheme of the direct redox (**A**) and electrocatalytic (**B**) detection of transition metal dichalcogenides (TMDs) by the nanoimpact method using screen-printed electrodes (SPEs). (**C**) Chronoamperometric responses illustrating the current spikes when a nanoimpact of TMDs particles takes place on the electrode surface. Adapted with permission from [130] Copyright 2015 American Chemical Society.

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
