# Peer review of "Electrochemical Detection and Characterization of Nanoparticles with Printed Devices"

_biosensors, 2019, doi:10.3390/bios9020047_

Round 1
Reviewer 1 Report
The manuscript is a review paper on the electrochemical techniques for the detection of nanoparticles. The manuscript is well structured, in good language, although minor grammatical errors and typos exist, and provides good explanations on the electroanaytical techniques that can be used for nanoparticle detection.
However:
The title of the paper is misleading since in the manuscript there is nothing about the printed devices that can be used for such detection. The author should include information not only on the sensing techniques but also on the printed devices. Otherwise, the title should be modified to more accurately represent the contents of the manuscript.
The author needs to modify the text, especially in the introduction,as to explain what the real advantages of nanoparticle sensing are. Currently, the manuscript gives reasoning on why nanoparticles are important and how they are used in biosensors for either amplification or indirect sensing. In contrast, what is really needed is to reason the sensing of nanoparticles and not how nanoparticles are used to improve the performance of other biosensors i.e. biosensors sensing other analytes with the use of nanoparticles.
In multiple places in the manuscript, the way the content is expressed, it confuses the user as to whether the paper is about the detection of nanoparticles or the use of nanoparticles to improve the performance of biosensors for other analytes.
The author provides explanations in simple terms, of the electroanalytical techniques used to detect nanoparticles. It would be much better and easier for the reader to understand if a figure illustrating, at least most of the papers included in the manuscript, the sensing/biosensing variation of each of the 5 techniques explained.
Author Response
I have attached a file with the response to the reviewer's comments.

Reviewer 2 Report
In this review paper, the recent advances of electrochemical strategies for the detection of nanoparticles using cost-effective printed devices are described.
This is an excellent review, well written and related to an interesting topic. I believe this manuscript can be accepted for publications in Biosensors but just minor revisions are necessary before publishing.
Please see my comments for more details.
1) Figure 1: The fonts are not consistent. I recommend revising the font/style of the middle writing “electrochemical detection of nanoparticles with printed devises ”, and use less animation.
2) Figure 1 and Figure 10: It is better to use the same style for the reaction. e.g A converts to B, or R converts to P.
3) Figure 10: This figure is similar to Figure 1 and 2. I recommend adding the corresponding electrochemical signals regarding each detection strategies. This makes the figure more clear and comprehensive, as well as making it less similar to Figure 1 and 2.
4) Line 70-71: Still authors need to expand the following sentence in the Introduction:
Consequently, screen-printed or paper-based electrodes have been widely employed as transducers for biosensing devices [23–25].
I recommend authors also add the recent trend of tattoo sensors, the wearable version of screen printed sensors; as they have very recently got attention in prestigious journals of Chemical Reviews, and Accounts of Chemical Research (below review papers). The authors can mention “The elegant and attractive application of such screen-printed sensors have also been demonstrated by fabricating body worn tattoo sensors for biosensing in diverse fields of medical (x), security (xx) and fitness (xxx)”.
I recommend citing the below papers by the pioneers of this field, Prof. John Rogers. These recent papers summarizes the elegant and advanced ways of employing screen printed sensors for biosensing.
x: Wearable Bioelectronics: Enzyme-Based Body-Worn Electronic Devices,
Acc. Chem. Res., 2018, 51 (11), pp 2820–2828
x: Bio-Integrated Wearable Systems: A Comprehensive Review, Chem. Rev. DOI: 10.1021/acs.chemrev.8b00573
x: Simultaneous Monitoring of Sweat and Interstitial Fluid Using a Single Wearable Biosensor Platform, Adv. Sci. 5 (2018), DOI: 10.1002/advs.201800880.
xx: Wearable potentiometric tattoo biosensor for on-body detection of G-typenerve agents simulants, Sensors & Actuators: B. Chemical 273 (2018) 966–972.
xxx: Skin-worn Soft Microfluidic Potentiometric Detection System, Electroanalysis. 31 (2019), 239-245.
5) Section 2.1 Direct detection methods, the below lines 123-126:
“Since a direct electron transfer needs to be produced between the electrode surface and the nanoparticles, they have to be at a very close distance to enable electron tunneling. This means that particles not close enough to the surface would not be able to be detected, making more difficult to get high sensitivity measurements.”
The effect of distance on electron transfer between the nanoparticle and the electrode surface area is comprehensively studied using electrochemical systems by Gooding et al. Then I highly recommend citing the following papers.
- Distance-Dependent Electron Transfer at Passivated Electrodes Decorated by Gold Nanoparticles, Anal. Chem., 2013, 85 (2), pp 1073–1080.
- The Influence of Organic-Film Morphology on the EfficientElectron Transfer at Passivated Polymer-ModifiedElectrodes to which Nanoparticles are Attached, ChemPhysChem2013,14, 2190 – 2197.
- An antifouling electrode based on electrode–organic layer–nanoparticle constructs: Electrodeposited organic layers versus self-assembled monolayers, Journal of Electroanalytical Chemistry 779 (2016) 229–235.
6) Figure labeling should follow the same style: A), B), C), or a, b, c…

Author Response

(The authors gave the same response as above.)

Reviewer 3 Report
The author reviewed several electrochemical detection methods for nanoparticles using printed electrodes. The manuscript highlights advances of electrochemical strategies and the examples of different techniques incorporated with the printed electrodes for nanoparticle detection. In general, the topic of the manuscript is interesting and appropriate to the review for Biosensors. However, the contents in the manuscript should be more systematic and informative. There are several specific major/minor changes I would recommend and the following items need to be explained and corrected in a revision.
First of all, it is likely that the authors just wrote down each paragraph based on just one or two papers and put them together without good interconnection. This is unacceptable for a review paper. The author should work on revising the manuscript to have a good flow.
In page 1, lines 9 and 11, the author would use different sequence works, instead of repeating the similar words. Revise the sentence with removal of either “On the one hand” or “On the other hand.”
In page 1, line 14, suggest revising the sentence, “In this review, I will introduce…” and avoiding the first person voice (I, me, mine), but use the passive voice in scientific writing.
In page 1, line 31, some critical literatures related to nanomaterials (e.g. peptide nanostructures, graphitic carbon nanomaterials) used for electrochemical biosensors are missed. They should be introduced to develop the solid rationale and contribution of the present study. The suggested articles are as follow: Biosens Bioelectron, 2012, 38, 295-301; Chem. Mater., 2013, 25, 2803-2811
In page 2, lines 52 and 59, suggest removing “very” It sounds subjective.
In page 2, line 86, revise the sentence to be a passive voice.
In page 2, line 87, the author should introduce types of printed devices or electrodes, as they are major contents in this review.
In page 3, line 103, the reasons to choose the strategies over other electrochemical techniques should be clearly stated. For instance, what are the advantages of the selected methods for nanoparticle detection, compared to other electrochemical techniques (e.g. EIS, electrochemical impedance spectroscopy, CV, ets.)?
In page 3, line 111, the author should provide an overview table in terms of different electrochemical techniques, target detection nanomaterials, quantitative values regarding detection performance, types of printed device and electrodes.
In page 4, line 145, how close the AuNPs contact to the electrode surface? Any references or quantitative data should be presented in the manuscript.
In page 11, lines 405-507, suggest revising the sentences to avoid using “in this case” repeatedly.
In page 15, line 591, “have” should be replaced with “has.”
In page 16, line 611, corrected as “because of”
In page 16, line 617, “I am sure that” sounds subjective. Revise the sentence to be a passive voice.
In page 16, line 621, suggest replacing “many” with “various”
Author Response

(The authors gave the same response as above.)

Round 2
Reviewer 1 Report
The author has revised the initially submitted manuscript in accordance to most of the reviewers' comments. Most of the comments from the reviewers have been fully addressed and the manuscript has been modified accordingly.
Specific comments:
The content of the paper has now been modified to include more information on the devices.
This comment has been partially addressed. The author has made the appropriate modifications that have clearly improved the manuscript. The comment is about the way the text is expressed and not so much on the content. It is understood that the relation of these devices with biosensor is very relevant to the readers of the journal and this relationship is correctly emphasized. The comment however, was about changing the expressions so that it will be clear to the reader that the devices describe detect nanoparticles either in solution or other mediums. Anyway, the manuscript is now clearer but could definitely be further improved
See the comments on point 2.
The comment addressed the fact that the author should explain the operation of each sensing technique as realized on the printed device through the use of a figure. More figures have been added and the manuscript is now in a much better condition, however more figures could help make the connection between the sensing principle and the devices clearer.
Author Response
A file with the response to the reviewer has been attached.
